# Nano-Immunomodulation: A New Strategy for Skeletal Muscle Diseases and Aging?

**DOI:** 10.3390/ijms24021175

**Published:** 2023-01-07

**Authors:** Francesco Millozzi, Andrea Papait, Marina Bouché, Ornella Parolini, Daniela Palacios

**Affiliations:** 1Department of Anatomical, Histological, Forensic Medicine and Orthopaedic Sciences, Section of Histology and Embryology, Sapienza University of Rome, 00161 Rome, Italy; 2IRCCS Fondazione Santa Lucia, Via del Fosso di Fiorano, 64, 00143 Rome, Italy; 3Department of Life Sciences and Public Health, Università Cattolica del Sacro Cuore, Largo Vito, 1, 00168 Rome, Italy; 4IRCCS Fondazione Policlinico Universitario A. Gemelli IRCCS, Largo Vito, 1, 00168 Rome, Italy

**Keywords:** skeletal muscle, inflammation, muscular dystrophies, sarcopenia, nanotechnology, immune modulation

## Abstract

The skeletal muscle has a very remarkable ability to regenerate upon injury under physiological conditions; however, this regenerative capacity is strongly diminished in physio-pathological conditions, such as those present in diseased or aged muscles. Many muscular dystrophies (MDs) are characterized by aberrant inflammation due to the deregulation of both the lymphoid and myeloid cell populations and the production of pro-inflammatory cytokines. Pathological inflammation is also observed in old muscles due to a systemic change in the immune system, known as “inflammaging”. Immunomodulation represents, therefore, a promising therapeutic opportunity for different skeletal muscle conditions. However, the use of immunomodulatory drugs in the clinics presents several caveats, including their low stability in vivo, the need for high doses to obtain therapeutically relevant effects, and the presence of strong side effects. Within this context, the emerging field of nanomedicine provides the powerful tools needed to control the immune response. Nano-scale materials are currently being explored as biocarriers to release immunomodulatory agents in the damaged tissues, allowing therapeutic doses with limited off-target effects. In addition, the intrinsic immunomodulatory properties of some nanomaterials offer further opportunities for intervention that still need to be systematically explored. Here we exhaustively review the state-of-the-art regarding the use of nano-sized materials to modulate the aberrant immune response that characterizes some physio-pathological muscle conditions, such as MDs or sarcopenia (the age-dependent loss of muscle mass). Based on our learnings from cancer and immune tolerance induction, we also discuss further opportunities, challenges, and limitations of the emerging field of nano-immunomodulation.

## 1. Introduction

The inflammatory infiltrate plays an important role in skeletal muscle regeneration, both in resolving the necrosis due to tissue damage, and in activating the repair processes [1]. The early infiltration phase is characterized by neutrophils, which arrive at the injured tissue after muscle damage. Neutrophils release free radicals, proteases, and chemotactic factors, such as cytokines, which activate monocytes and macrophages able to remove the disrupted myofilaments, other cytosolic structures, and the damaged fibers. The first pro-inflammatory phase, characterized by the presence of M1 macrophages, is then followed by a secondary anti-inflammatory phase characterized by the polarization of M1 to M2 macrophages. The M2 macrophages, through the production of anti-inflammatory cytokines such as IL-4, IL10, and IL-6, contribute to the resolution of the inflammatory response required for skeletal muscle repair [2]. (Figure 1).

As it occurs with other degenerative diseases, many muscular dystrophies (MDs) present an altered immunological response, which contributes to the pathogenesis of the disease [3,4,5]. Consistently, treatment with corticosteroids, such as prednisone or deflazacort, to palliate the disease symptoms is the standard-of-care treatment for some MDs, including Duchenne muscular dystrophy (DMD), facioscapulohumeral dystrophy (FSHD), and congenital muscular dystrophies (CMDs) [6,7,8,9]. Immunomodulation is also a relevant therapeutic approach for other pathological skeletal muscle conditions, such as sarcopenia, or the loss of muscle mass that occurs with age. Sarcopenia is associated with changes in the skeletal muscle microenvironment, such as an imbalanced inflammatory infiltrate and high levels of pro-inflammatory cytokines [10,11].

Several strategies are being investigated to restore a proper immune balance in diseased muscles using cytokines and other immunomodulatory agents. However, many of them have failed due to poor pharmacokinetics and undesired pleiotropic effects. In this context, nanomedicine offers a plethora of unprecedented tools to increase drug stability, improve selectivity, and reduce toxic effects. Nanomedicine employs nanoscale materials as drug delivery systems (DDSs) by exploiting the fact that endogenous transport at the cellular level is actively driven at the nanometer length scale [12]. The high surface-to-volume ratio of nanoparticles (NPs) facilitates the loading of cytokines [13], nucleic acids [14], and other bioactive molecules. In addition, the use of different surface chemistries allows us to functionalize the NPs with different targeting moieties, a characteristic that can be exploited to favor selective targeting. In addition, some nanomaterials have intrinsic immunomodulatory properties, further expanding the potential of nanotechnologies as immunomodulators [15]. However, despite being extensively explored in cancer and immune tolerance induction, the use of nanotechnologies for immunomodulation in regenerative medicine is still in its infancy. Interestingly, the first works are now emerging and are very promising. Here we review the current state-of-the-art in the field, highlighting the potential of nano-immunomodulation in the treatment of muscle diseases.

## 2. The Immune System in Skeletal Muscle Regeneration and Aging

The skeletal muscle, which in physiological conditions is characterized by a small turnover of multi-nucleated myofibers, has a remarkable capacity to regenerate after damage. The main effectors of muscle regeneration are satellite cells, a population of quiescent, muscle-resident stem cells initially identified by their anatomical location beneath the fiber lamina [16]. After an injury, that can be either physical or genetic -as occurs in many MDs- satellite cells exit from the quiescence, start to proliferate, and migrate to the site of the lesion, where they fuse to repair the damaged fibers. However, not all the activated satellite cells continue to proliferate, and a small subset of them re-enter the quiescence to maintain the stem cell pool required for successive rounds of regeneration. The satellite cells’ function is determined by complex regulatory mechanisms involving intrinsic and extrinsic factors, as well as by their interaction with other cell populations present in the regenerative microenvironment, such as fibro-adipogenic progenitors (FAPs) or cells from the inflammatory infiltrate [17,18].

As a consequence of tissue damage, there is a strong recall of immune cells with the extravasation of those playing a cleansing role, removing dead cells, and phagocytosing any bacteria that have entered, consequently stimulating the processes of repair, regeneration, and regrowth of damaged tissue [5,19,20]. In fact, the regenerative potential of a skeletal muscle is closely related to its interactions with the immune system. Initially, there is a strong recruitment of cells belonging to the myeloid lineage, with only a small proportion of lymphoid cells being recruited [21]. The first cells to be recruited are represented by neutrophils, which, when extravasated, produce and release ROS, perform a cleaning action from cell debris by removing the dead cells, and release pro-inflammatory cytokines and chemokines. This consequently results in a recall action for other immune cells, such as macrophages [22] and eosinophils [23]. Neutrophil depletion in murine models via acute toxin-mediated injury has been shown to delay muscle regeneration and alter the kinetics of the immune response [24]. However, despite their key role in the early phases of regeneration, their prolonged permanence at the site of the lesion results in increased damage. In fact, many studies have reported how neutrophil depletion in several models of muscle damage, such as ischemia followed by reperfusion, or of exhaustive exercise or lengthening contractions, can result in positive effects by reducing muscle tissue degeneration [25,26,27].

Macrophages are the most abundant inflammatory cells in injured muscles, which persist in the muscle for many weeks after the regenerative process. Some of them may perform a similar action as neutrophils by releasing free radicals [28,29]. In addition, they also exert debridement actions and release chemokines and immunoregulatory factors. These, on one hand, amplify the inflammatory response [30,31,32] and, on the other, regulate the function of different cell populations present in the damaged tissue [33,34,35,36,37,38]. 

Macrophages can be broadly divided into two different populations: the M1 and the M2 macrophages [39]. M1 macrophages are characterized by an elevated ability to secrete pro-inflammatory cytokines such as IL-1β, TNF, IL-12, and IL-18, and contribute to supporting the inflammatory response [40]. M2 macrophages, on the contrary, are characterized by the expression of the macrophage mannose receptor (MMR), also called the CD206 and of CD163 [41,42], and can be found in different states. Relevant to muscle regeneration are the alternative activation state M2a and the anti-inflammatory state M2c [39]. M2 macrophages play a key role in both reducing the pro-inflammatory response and in promoting muscle regeneration through the activation of satellite cells. These effects are mediated by the release of cytokines and growth factors, such as IL-10, Insulin-like growth factor 1 (IGF-1), and Klotho [38,40,43,44]. During tissue remodeling, the macrophages undergo a differentiation continuum from pro-inflammatory M1 macrophages to immunoregulatory M2 macrophages, playing a fundamental role in the regeneration process [45,46]. Indeed, experiments in which macrophage depletion was performed resulted in a strong delay in reparative processes [47]. This was also observed in models of cardiotoxin-induced damage. Again, the depletion of the CD11b cells, largely represented by macrophages, resulted in a strong reduction in regenerative potential, with an increase in residual necrotic fibers and fat accumulation [37,48,49].

The role of lymphoid populations in the regeneration process has only recently gained the attention of researchers, mainly due to the low presence of these cells in damaged tissues [50]. Support for a possible pro-regenerative role for these populations, however, comes from T or B cell ablation experiments, in which a strong delay or reduction in regenerative capacity was reported [51,52]. In particular, it was observed that the reduction in the presence of pro-inflammatory cytokines such as IL-1α, TNFα, and IFNγ, cytokines typically released by T lymphocytes, underlies the reduction in myogenic cell proliferation [51]. In addition, depletion of the cytotoxic CD8α population delays muscle regeneration [52]. CD8KO mice have lower amounts of satellite cells, smaller regenerative myofibers, and they show increased fibrosis two weeks after injury [53]. This effect appears to be related to a strong reduction in the expression of the chemokine CCL2, which plays a key recruitment role for myeloid cells [52,54]. On the other hand, CD4 lymphocytes, and in particular a special population of Foxp3+ CD4 regulatory T (treg) cells, are also recruited to the site of the lesion following acute injury [55,56,57,58]. These cells are known for their immunoregulatory action due to their ability to release IL-10, TGFβ, and amphiregulin, which has been shown to stimulate the satellite cells’ differentiation without impacting their proliferation [55]. Consistently, the depletion of these cells has been shown to reduce the rate of regeneration in a manner comparable to the macrophage depletion [47].

During aging, there is a change in the immune system called inflammaging, a condition in which the system shifts toward a stronger inflammatory response characterized by high levels of inflammatory cytokines typically released by Th1-type lymphocytes. Consistently, aged muscles already exhibit elevated levels of pro-inflammatory cytokines, such as IL-1β, IFNγ, and TNFα [59,60]. At the same time, an increase in M2 polarization is observed within the muscle, as shown by an increase in the number of double-positive CD163 and CD206 M2a macrophages. The age-related increase in M2a pro-fibrotic macrophages and the associated muscle fibrosis were shown to depend, in part, on the age of the bone marrow cells [61]. This phenomenon may also be partly explained by the increased level of IL-10 observed during muscle aging, and could represent an intrinsic compensatory mechanism in response to the systemic changes due to inflammation [61]. Interestingly, Sloboda and colleagues reported lower IL-10 levels upon injury in the muscle of aged mice as compared to young mice. The authors supposed that this effect can be due to the reduced ability in M2 aged macrophages to produce immunoregulatory cytokines during the regeneration process [62].

Since macrophages exhibit a dynamic and plastic pattern presenting both typical features of inflammatory and anti-inflammatory macrophages, depending on the stimuli, the changes in the muscle microenvironment during aging alter the macrophages’ phenotype and function [63]. For instance, IGF-1 plays a key role in muscle regeneration, and its overexpression has been reported to underlie inflammation-resolving processes due to its ability to induce the differentiation of anti-inflammatory macrophages [43]. However, its production decreases with age [64]. Similar to IGF-1, other factors known to promote regeneration, such as Klotho, are down-regulated in old muscles, and this contributes to the impaired muscle regeneration that occurs with age [65]. On the other hand, the increased levels of circulating IL-6 in aging might contribute to the decline in skeletal muscle function [66,67,68,69].

The inflammatory response that occurs with age is also exacerbated by a reduction in the polarization of CD4 T lymphocytes towards treg cells. This appears to be related to an impairment in the production of IL-33, a cytokine released from the FAPs and observed to drop dramatically in aged mice [70]. As one of the main functions of the treg cells during skeletal muscle regeneration regards the regulation of macrophage polarization, the age-related reduction of this population could be involved in chronic pro-inflammatory signaling and diminished tissue regeneration.

The imbalance between pro-inflammatory and anti-inflammatory signaling is associated with an impairment of the satellite cell function in old muscles [71]. This is further supported by experimental evidence showing that transplantation of old bone marrow cells into young animals reduces satellite cell numbers, and promotes their switch toward a fibro-adipogenic phenotype [72]. Recent studies point out macrophage-released TNFα as a key mediator of this effect [73]. Consistent with these observations, aberrant activation of the TNFα downstream target NF-κB in old muscles impairs the funcyion of the satellite cells function and delays regeneration [74].

## 3. The Burden of the Immune Response in Muscle Disorders

Whereas under physiological circumstances, inflammation is a fundamental part of the pro-regenerative response, aberrant inflammation is a hallmark of many MDs, including dystrophin-deficient muscular dystrophies (Duchenne and Becker muscular dystrophies, DMD and BMD), congenital muscular dystrophies, dystroglycanopathies, and FSHD. In MDs, myofiber instability leads to chronic inflammation, which then contributes to the pathogenesis of the disease by impairing regeneration and inducing fibrosis [3,5,8,75]. It is therefore not surprising that immunomodulators are extensively used in clinics for the treatment of many MDs [6,7,8,9,76].

The impact of the immune system on the progression of MDs has been extensively studied in a murine model of DMD, the mdx mice. Local and systemic inflammation are associated with both muscle degeneration and fibrotic deposition in mdx mice, through different mechanisms [39,75,77]. From a cellular point of view, a deregulation of both lymphoid and myeloid functions has been observed. Different from what was observed in acute damage, the T cells are among the first cells recruited in chronic lesions, and they play a key role in regulating the inflammatory response [4]. The depletion of CD4 T helper lymphocytes as well as CD8 cytotoxic lymphocytes in mdx mice reduced the amount of muscle damage [78]. On the contrary, the depletion of treg cells is associated with exacerbation of muscle damage, increased IFNγ release, and increased M1 inflammatory macrophage response [58]. Similar effects were observed in mdx mice in which IL-10 ablation was performed. In this model, increased muscle damage and reduced strength were observed. Macrophages isolated from these mice showed a distinct M1 phenotype with elevated iNOS marker expression and increased cytotoxic activity compared with macrophages isolated from wild-type controls. This effect was attributable to an imbalance in the macrophage immune response. In fact, ablation of the IL-10 resulted in a reduced bias to M2c anti-inflammatory macrophages. The authors also reported that regenerating myofibers in mdx mice express the IL-10 receptor, thus suggesting that this cytokine could also have a direct effect on muscle cells [79]. Finally, macrophage depletion in a mouse model of DMD leads to adipogenic conversion of the cells and exhaustion of the stem cell pool [37]. To further complicate the puzzle, it has been recently demonstrated that the spleen is the dominant reservoir of pro-inflammatory monocytes in mdx, and that splenic monocytes play a critical role in both muscle fiber injury and repair, different from the bone marrow-derived monocytes [80]. Splenectomy performed before disease onset significantly reduced the number of pro-inflammatory monocytes infiltrating the dystrophic limb muscle, resulting in a significant reduction in inflammation and necrosis, along with improved regeneration during early disease. However, during late disease, the lack of splenic monocytes adversely affected muscle fiber repair due to a delay in the phenotypic shift of pro-inflammatory M1 to anti-inflammatory M2 macrophages, which is not compensated by bone marrow-derived monocytes [80].

The pro-inflammatory cytokines present in the dystrophic muscle ultimately promote M1 macrophage activation, resulting in a persistent inflammatory response. On the other hand, the decrease in anti-inflammatory cytokines blocks M2-macrophage expansion and enhances oxidative stress, TGF-β secretion, and the expression of fibrotic genes [39]. In this context, pharmacological inhibition of pro-inflammatory cytokines represents a promising therapeutic strategy for DMD patients.

Inflammation has also been described as part of the pathogenic mechanisms in other MDs. CMDs, Emery Dreifuss Muscular Dystrophy (EDMD), and LGMD are considered inflammatory diseases [81,82]. In DyW mice, a murine model of merosin-deficient CMD, an aberrant inflammatory response with high levels of infiltrating macrophages and pro-inflammatory cytokines were observed. This is accompanied by an increase in NF-kB signaling [83]. Activation of NF-kB has also been observed in laminA/C laminopathies, where mutations in the lamin A/C led to structural alterations in the nuclear lamin of dystrophic macrophages [84]. This correlates with an up-regulation of toll-like receptor s(TLR) and aberrant levels of pro-inflammatory cytokines, such as IL-6 or IL-8. α-Sarcoglycan-deficient mice also present high levels of pro-inflammatory cytokines, such as IL-1β, IFNγ, and IL-6, and increased the CD45 and CD4 infiltration [85]. Finally, in dysferlinopathies, muscle inflammation due to the activation of innate immune receptors such as TLRs has been observed. Consistently, the mice lacking both dysferlin and myeloid differentiation primary response gene 88 (*MyD88*), a key mediator of the TLR-dependent innate immune signaling, exhibit improved regeneration and increased muscle force [86]. This observation agrees with previous work showing the involvement of TLRs-mediated signaling, NF-kB, and the assembly of the inflammasome NLRP3 in the pathogenesis of dysferlin-lacking mice [87].

Taken together, all this evidence suggests that, in addition to restoring the genetic defect, a benefit could also be achieved by restoration of a correct immune balance. The re-establishment of proper immune balance would in fact lead to a reduction of inflammatory cells with a consequent reduction of damage, stimulation of regenerative processes, and ultimately restoration of muscle function.

## 4. The Regulation of Immunity in Muscle Disorders

As mentioned above, the standard of care treatment for many MDs, including DMD, is based on immunomodulation and, in particular, on the application of steroids. Steroids allow broad-spectrum modulation of the immune response in order to stimulate regenerative processes or at least reduce tissue damage. In fact, it has been seen that the use of these pharmacotherapies can promote a recovery of muscle strength while maintaining and preserving the muscle mass itself. On the other hand, there are a great number of side effects, such as weight gain and osteoporosis gastric issues, that have greatly limited its use even in the treatment of DMD patients [6,7,8,9].

For these reasons, some attempts to obtain targeted modulation of the immune response have been made. To restore the immune balance, the pro-inflammatory response must be counterbalanced by an immunoregulatory response. This means that the polarization of the macrophages into M1-type macrophages must be pushed toward the induction of type 2 macrophages. The same goes for the polarization of T lymphocytes, which from inflammatory Th subsets (Th1, Th17) must be directed toward the production of cells with strong immunoregulatory action such as treg [75,88].

Several strategies have been employed in this regard. For instance, treatment with prednisone in DMD patients was observed to induce a shift from M1 to M2 macrophages and a reduction of autoreactive T lymphocytes. On the other hand, the modulation of pro-inflammatory cytokines showed promising results. In particular, the use of neutralizing antibodies against the TNFα receptor led to a reduction in fibrosis deposition and necrosis in mdx mice [89,90]. On the contrary, the administration of anti-inflammatory cytokines, including IL-4 and IL-10, exerts an immunoregulatory action by stimulating a bias toward treg-like cells and M2 macrophages. This effect has been reported in several disease models, including in neurodegenerative disorders such as demyelinating diseases, arthritis, and psoriasis [91,92]. Interestingly, local delivery of IL-10, injected at early time points after cardiotoxin-induced muscle damage, when pro-inflammatory cytokine expression is predominant, induced premature differentiation of the satellite cells, which in turn resulted in the formation of smaller muscle fibers. This effect could be reversed by the concomitant administration of TNFα [93]. However, the use of cytokines raises numerous issues in terms of pharmacokinetics, with the requirement for repeated administration as well as the difficulty of being able to control their pleiotropic effect, especially in molecules that may also target different tissues [94]. For example, in the case of IL-10, it is sadly known how many clinical trials have failed precisely because of its instability in vivo [95]. To overcome this problem, several strategies have been attempted, including interleukin PEGylation to protect them from degradation [96].

Another important target is the transcription factor NF-κB, which in DMD has been shown to be highly expressed in both the immune and muscle cells, making it a major target for drug therapy [97]. However, the results obtained using pharmacological inhibition were somehow controversial. In fact, NF-κB blockade during the inflammatory peak was able to reduce the inflammatory response itself, while its inhibition during the resolving/regenerative phase was involved in prolonging the inflammatory response [98]. In the same direction goes TGFβ. Indeed, it has been reported that this growth factor has both anti- and pro-inflammatory actions, but the mechanisms behind this ambivalent behavior are not yet fully understood. Early inhibition of TGFβ reduced fibrosis in mdx mice, while increasing the number of pro-inflammatory CD4 lymphocytes with a Th1 phenotype [99]. In contrast, it was recently observed that the increased TGFβ levels present in the mdx muscles lead to an expansion of FAPs, resulting in an impairment of myogenesis. Consequently, the TGFβ inhibition reduces the FAPs accumulation and could result in a beneficial effect [100].

Other immunomodulatory molecules include rapamycin, whose administration results in a strong reduction in the T-type inflammatory infiltrate (CD4 and CD8 cells) while increasing the amount of treg cells [101]. This is consistent with what was observed in other disease models in which rapamycin, acting on the mTOR pathway, stimulates the polarization, expansion, and survival of the treg cells [102,103]. Alternatively, the use of IL-2 blockers was observed to induce improvement by reducing creatine kinase release, improving muscle histology and cytoarchitecture, and again increasing the amount of treg cells [58]. In this context, the inhibition of PKCθ, a critical regulator of the effector T-cell activation, whose blockade enhances the treg function [104], has been shown to markedly improve the disease pathology by reducing the size and altering the quality of the inflammatory cell infiltrate in the dystrophic muscle [4,105].

Immunoregulatory strategies have been studied in other forms of MDs, such as dysferlinopathies. Consistently with what was observed in DMD patients, pharmacological treatment to block TNFα showed reduction in inflammation, necrosis, and fibrosis in dysferlinophatic SJL/J mice [106]. In addition, the administration of halofuginine, -a T helper cell inhibitor improves dystrophic features in a dysferlin-deficient mouse model [107].

In summary, the possibility of being able to control the inflammatory process by regulating the response of innate and adaptive immunity can be used to delay or counteract the progression of MDs. In fact, the restoration of the immunological balance can lead to an increase in the muscle function, triggering the regenerative process and consequently counteracting the negative effects induced by the chronic inflammatory process.

## 5. Nanomedicine-Based Strategies for Immunomodulation

As discussed above, despite their extended use in the clinical practice, immunomodulatory agents have considerable adverse effects when given at therapeutically relevant doses. Therefore, strategies aimed at reducing the off-target effects by lowering the effective dose or by increasing tissue and/or target selectivity emerge as relevant clinical opportunities. Within this context, different types of nano-sized platforms provide a viable strategy for immune modulation. Based on their composition, NPs are divided into organic (i.e., liposomes, polymers, solid lipid NPs), and inorganic (i.e., metal, oxides, carbon-based, silica, etc.), with different safety profiles and immunomodulatory capacities [108,109,110]. Organic NPs have a long clinical history and can guarantee biocompatibility and biodegradability. Inorganic NPs present a higher chemical stability, and are easy to synthetize and functionalize. In addition, they are responsive to both internal (pH, temperature, redox potential) and external (light, ultrasound and magnetic field) stimuli. Furthermore, the unique optical properties (fluorescence, plasmonic absorbance, etc.) of these NPs allow us to obtain precise spatiotemporal control (reviewed in [111]). However, despite these attractive properties, inorganic NPs are significantly less mature in terms of clinical translations and their potential toxicity is a significant matter of concern [112].

Based on their chemical and physical parameters, the different types of NPs exhibit different behaviors. For example, liposomal NPs received much attention in the cancer field due to their ability to deliver immunomodulatory agents [113]. In this context, previous work showed the ability of liposomal doxorubicin NPs to increase the concentration of therapeutic drug on tumor-associated macrophages in comparison to delivery of free drugs [114]. On the other hand, several studies using single-walled carbon nanotubes for biomedical applications showed their high immunotoxicity [115]. In addition, citrate coated supermagnetic iron oxide NPs were used as immunotherapic magnetic drug delivery system to target “cold” tumor [116].

To date, most of the nanotechnologies applied to immune modulation are focused on tolerance induction [117,118] and cancer therapy [119,120], and therefore have antigen presenting cells (APCs) as their main target. Nanocarriers represent an excellent strategy to present allogeneic antigens or to conjugate molecules capable of modulating dendritic cell activation, as well as the function of other components of both innate and adaptive immunity [121,122]. Several NPs have been designed to inhibit monocyte production for the treatment of inflammatory diseases through the delivery of small interference RNAs (siRNAs) and small molecules. For instance, it has been observed that the CCR2 small interfering RNA (siRNA)-loaded NPs reduced the number of monocytes accumulated in sites of inflammation and suppressed the progression of inflammatory diseases, such as atherosclerosis, myocardial infarction, and pancreatic islet transplantation in diabetes [123]. A similar strategy using the MCP-1 siRNA-loaded lipid–polymer NPs successfully inhibited the mobilization and recruitment of inflammatory monocytes to the diseased heart from haematopoietic niche in a mouse model of myocardial infarction [124]. Finally, PLGA nanoparticles loaded with peroxisome proliferator-activated receptor (PPAR)γ agonist Irbesantan blocked the inflammatory monocyte infiltrate in a mouse model of myocardial ischemia–reperfusion (IR) injury [125].

Only recently, several groups started to use different types of NPs to modulate the immune response in regenerative medicine, with some examples already emerging in the field of skeletal muscle regeneration (Figure 2). Conjugating the immunomodulatory agent to different types of nanostructures allows therapeutic levels of the drugs while reducing toxic effects. For instance, PEG-stabilized nano-liposomes were used to deliver the steroid pro-drug methylprednisolone in a mouse model of the DMD [126]. The efficacy and safety of the treatment efficacy was shown by the reduced inflammation and long-term improvement in muscle function. In addition, some nano-sized materials have intrinsic immunomodulatory properties, further expanding the potential of nanotechnologies for the treatment of muscle diseases.

### 5.1. Nanocarriers to Deliver Immunomodulatory Agents into Diseased and Aged Muscles

In the context of muscular diseases, the main strategies explored so far aim at promoting macrophage polarization from a pro-inflammatory M1 to a pro-regenerative M2 subtype, and increasing the T cells’ tolerance. For example, in a study by Raimondo and colleagues, it was seen that the PEGylated gold NPs loaded with IL-4 induce macrophage polarization toward M2a-type macrophages when injected intra-muscularly in a mouse model of ischemic injury. This was reflected in an amelioration of muscle damage [127]. A follow-up study by the same authors also showed that gold NPs conjugated with IL-4 or IL-10 were able to induce an increase in muscle strength and regeneration due to the reduction of the inflammatory process as a direct consequence of the reduction of cytotoxic T cells and increase in treg cells [127]. This confirms what was previously reported in another study, where IL-4 administration was responsible for the induction of treg cells [128]. Other than gold NPs, IL-4 has also been conjugated to mesoporous silica NPs, again triggering macrophage polarization in vivo [129].

A different strategy to modulate the levels of the anti-inflammatory cytokines relies on the delivery of plasmids or small non-coding RNAs (i.e., microRNAs). The CD44-targeting hyaluronic acid-poly (ethyleneimine) NPs (HA-PEI) containing plasmids expressing IL-4/IL-10 or microRNA-223 (a potent regulator of inflammatory responses through down-regulation of several genes such as *HIF-1α*, *PPARγ* and *STAT3* [130,131,132] have been applied to modulate macrophage reprogramming into the M2 subtype in wild type of mice, and could represent a potential strategy also for muscles diseases [133,134].

In addition to their potential to tamper the aberrant immune response in some MDs, NPs-mediated delivery of immunomodulators is also showing promises in mitigating vector immunogenicity in gene therapy applications. Currently, there are several approved clinical trials aimed at assessing the safety, biological activity, and efficacy of delivering a functional copy of the dystrophin gene to DMD patients (clinicaltrials.gov, 06-11-2022). All the protocols are based on the use of non-replicating recombinant adeno-associated viruses (AAVs) for gene delivery into dystrophic muscles [135,136]. However, AAVs are highly immunogenic in humans, which impairs a second administration of the therapeutic vector if needed [135]. In addition, a significant proportion of the population has pre-existing resistance to AAVs [137], making those children ineligible for the treatment. In a seminal work by Meliani and co-workers, co-treatment with rapamycin-loaded PLA NPs was shown to tamper AAV-mediated immunogenicity both in mice and non-human primates, allowing re-treatment with the vector [138]. These experiments opened the door to the possibility of combining gene therapy with nano-immunomodulation in the treatment of MDs [111]. Further work will assess if such a strategy is useful for tampering with AAV immunogenicity in this and other gene therapy applications.

All these results support the use of different nanocarriers for the delivery of therapeutic bioactive compounds to modulate the immune response in diseased and aging muscles. In addition, many NPs have intrinsic immunomodulatory properties, or can be designed to modulate their interaction with the immune system, as discussed in the following sections.

### 5.2. Nanomaterials with Intrinsic Immunomodulatory Properties

Different nanosized materials are capable of regulating the immune response to some extent based on their size, composition, and inoculation site [139,140,141]. 

For instance, many inorganic NPs such as silver, gold, titanium oxide, and cerium oxide NPs may act as immunomodulators [15]. Some studies have reported that both silver and gold-silver NPs impact macrophage polarization, albeit with different efficiencies. Gold-NPs seem to be more efficient than silver NPs in up-regulating pro-inflammatory genes, and this effect is dependent on the different uptake routes [142]. In a different study, the pro-inflammatory effects of silver, aluminum, carbon black, carbon-coated silver, and gold NPs on murine macrophages were compared and related to the ability of the different NPs to trigger the activation of the NF-kB pathway [143]. Silica NPs have also shown regulatory effects on macrophages [144], suggesting that macrophage polarization by inorganic NPs is a generalized effect. However, it is important to point out that differences were observed across studies, mainly due to the different NP sizes, concentrations, and surface modifications [144,145,146,147,148].

This intrinsic immunomodulatory activity is not exclusive to inorganic NPs. For instance, NPs formed of poly(lactic acid) (PLA) with either poly(vinyl alcohol) (PVA) or poly(ethylene-alt-maleic acid) (PEMA) mitigate macrophage pro-inflammatory cytokine secretions induced by LPS through abrogation of NF-κB and p38 MAPK activation [149].

It has also been observed that NPs, regardless of their composition, may trigger an immune response, and this immunomodulatory ability is strictly dependent on their size. In fact, it has been reported that NPs with a diameter of 500 nm are mainly phagocytosed by macrophages located in the marginal region of the spleens through the MARCO scavenger receptor. This induces the activation of dendritic cells, which in turn triggers the polarization of CD4 lymphocytes towards the treg cells and, in parallel, induces their anergy [139]. In contrast, NPs smaller than 50 nm were rapidly delivered to the lymph nodes, where, being in direct contact with dendritic cells, they were readily able to deliver their biological effect by stimulating the induction of tolerance [140]. Other than size, NPs’ shape can also affect the immunological response. The most commonly used shapes are rods, spheres, and shells [150]. A study using rod, spherical, or cubic gold NPs coated with West Nile virus envelopes showed a different cytokine secretion in dendritic cells [151]. Furthermore, in another study, Chen et al., observed a higher increase in the TNFα and IL-8 secretion when they used short rod-shaped capsules as compared to spherical and long rod-shaped capsules [152]. Finally, it has been observed that spherical glyco-nanoparticles (GNPs) were more efficiently internalized in RAW 264.7 macrophages as compared to cylindrical GNPs. On the other hand, cylindrical GNPs induce a higher increase in IL-6 cytokine production than spherical GNPs [153]. All these results indicate the importance of the NPs’ shape and size in modulating immune responses.

As mentioned above, the intrinsic immunomodulatory activity of some NPs could have important implications when designing a nanomedicine-based approach in regenerative medicine, especially in those cases in which the protocol foresees a systemic delivery of the therapeutic NPs. The NPs’ behavior in vivo is intrinsically dependent on their interaction with the immune system. Once the therapeutic NPs reach the bloodstream, they are immediately coated by a dynamic protein layer called the “protein corona”. This protein corona significantly affects not only the NPs’ fate (pharmacokinetics, biodistribution, and target recognition), but also their effectiveness as therapeutic biocarriers and/or immunomodulators [154,155,156]. Interestingly, recent works have now opened the possibility to engineer the “protein corona” to modulate NP interaction with the immune system [157,158]. If this strategy could be used to improve NPs’ potential as immunomodulators deserves further investigation.

## 6. Conclusions and Future Directions

While nanotherapeutics for regenerative medicine, and in particular for skeletal muscle disorders, are still in their infancy, nano-immunomodulation has been actively explored in other fields such as cancer and immune tolerance induction [159,160]. The extensive work in these fields provided important pre-clinical evidence on how different types of nanocarriers can be used to modulate the immune response to obtain specific cellular responses.

However, when moving into the clinics, less than 15% of the tested nanotherapies for cancer successfully concluded phase III clinical trials [161]. The reasons for this overall failure come from the limited reliability of current animal models for pre-clinical studies and the difficulty of establishing scalable manufacturing processes for nanomedicines. Several parameters must therefore be carefully considered when moving nano-based drugs into clinics. These include, on the one hand, biological parameters such as efficiency, toxicity, and in vivo NP behavior and, on the other, manufacturing and regulatory issues, such as large-scale production and authorities’ approvals (reviewed in [111]).

While the efficiency of different organic and inorganic NPs as biocarriers can be improved by modulating their composition, shape, surface-to-volume ratio, or surface chemistry, toxicity issues play a key role in nanoplatform development and are overall less explored [162]. The fact that many promising solutions are based on inorganic materials poses additional concerns, and only a few inorganic nanomaterials have been approved by the regulatory agencies [161]. Understanding the in vivo behavior of these nanomaterials and their intrinsic interaction with the immune system, as we discussed here, will be fundamental to accelerating their transition to clinics.

To conclude, nano-immunomodulation emerges as an excellent tool for regenerative medicine. As extensively discussed in this review, aberrant immune responses are a hallmark of many degenerative diseases, including MDs. Therefore, immunomodulation has been extensively applied in clinics for the treatment of muscle disorders. However, this comes with important undesired effects, such as weight gain, mood changes, and an increased risk of infection. Nanotechnology is expected to provide unique features and new methodologies for immune modulation in regenerative medicine, increasing efficiency and reducing off-target effects. The challenge now is to develop smarter, multi-responsive materials with controllable and robust delivery that should be broadly biocompatible, but whose presentation to the immune system per se may prove capable of inducing the desired regulatory response. There is no doubt that the next few years will see an explosion of nano-based therapies for muscle disorders.

## Figures and Tables

**Figure 1 ijms-24-01175-f001:**
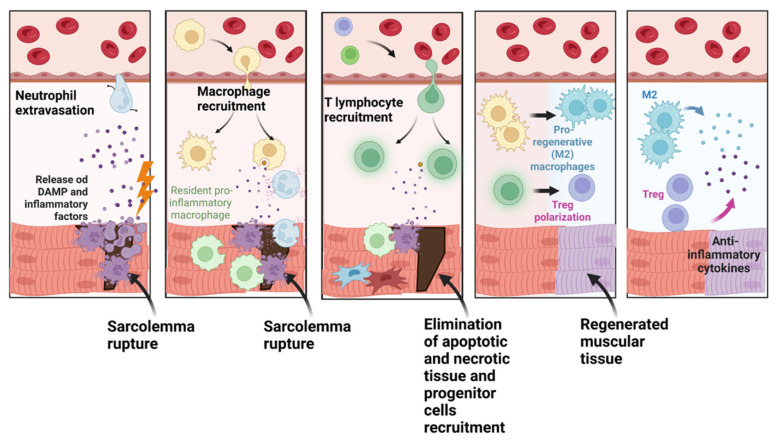
The evolution of the inflammatory response in muscle damage. When muscle tissue is damaged, the release of inflammatory factors and damage-associated molecular patterns (DAMPs) occurs, which attract various cells of the immune system. The first to intervene are the neutrophils, which extravasate to the site of injury. Here, in the case of sterile inflammation, i.e., in the absence of bacteria or other pathogens, they will start to perform a cleansing action against the apoptotic or damaged cells. In addition, they will produce reactive oxygen species (ROS), which will in turn be responsible for recruiting other types of immune cells. The inflammatory state induced by the damage also leads to the involvement of resident macrophages in the first instance, while others will be attracted by the chemotactic factors released by both the damaged muscle cells and neutrophils. Once the cleansing action of the apoptotic and necrotic cells is complete, the recall of T lymphocytes and muscle progenitors occurs. The latter, stimulated by the cascade of inflammatory signals, will be stimulated to repair the damage with the restoration of functional homeostasis. The repair process concludes with the polarization of T lymphocytes and macrophages towards immunoregulatory subsets (treg and M2 macrophages), which, through the release of cytokines and other factors, will modulate the immune response until the inflammatory status ceases. (Figure designed using Biorender.com, accessed on 7 November 2022).

**Figure 2 ijms-24-01175-f002:**
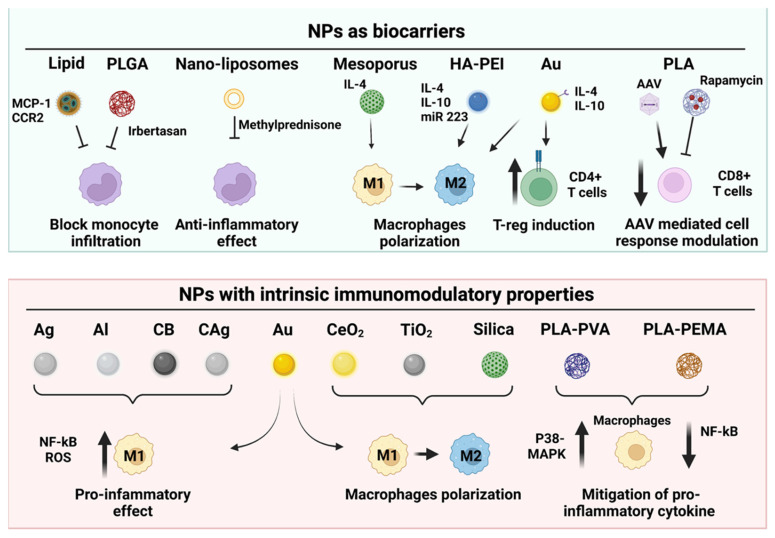
Nanoparticles-based strategies for immunomodulation in muscle diseases. Overview of different types of NPs used as biocarriers to deliver immunomodulatory agents and of NPs with the intrinsic ability to modulate the inflammatory response. (NP, nanoparticle; PLA, poly (lactic acid); PLGA, poly(lactic- co -glycolic acid); PVA, (poly(vinyl alcohol)); PEMA (poly(ethylene-alt-maleic acid)); HA-PEI, hyaluronic acid-poly ethyleneimine; AAV: Adeno-associated virus; Ag, Silver; Al, Aluminium; CB, Carbon black; CAg, Carbon-coated silver; Au, Gold; TiO2, Titanium oxide; CeO2, Cerium oxide. The mechanism of action on the different immune cell populations is also shown. (Figure designed using Biorender.com, ccessed on 7 November 2022).

## Data Availability

Data sharing is not applicable to this article as no new data were created or analyzed in this study.

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
