# Peer review of "Nano-Immunomodulation: A New Strategy for Skeletal Muscle Diseases and Aging?"

_ijms, 2023, doi:10.3390/ijms24021175_

Round 1
Reviewer 1 Report
The authors report a review of the possibility of modulating the immune response at the nanoscale in muscles to fight diseases and aging.
I suggest addressing the following points:
- Please check Figure 1, which is not well formatted.
- The part on the regulation of immunity in muscle disorders can be expanded with the discussion of further studies.
- A table with the current clinical trials of nanomedical treatment against muscle disorders (e.g., DMD) will be useful for the reader to underline the actual stage of the research.
- In the section "Nanomaterials with intrinsic immunomodulatory properties", please the authors mention and expand the discussion on the shape of the nanomaterials, a parameter that has a relevant role in the delivery and in the immune response.
Reviewer 2 Report
In this review, the authors have collected nano-sized materials for the modulation of aberrant immune response in some physio-pathological muscle conditions, including MDs and sarcopenia. The authors have also discussed opportunities, challenges and limitations of the emerging field of nano-immunomodulation. Overall, this review can inspire more material design ideas of nano-immunomodulation. Therefore, I would like to recommend this work to publish in International Journal of Molecular Sciences. Below are some comments for the authors.
1. Figure was cut off. Please provide complete graph.
2. In the section of “Nanomedicine-based strategies for immunomodulation”, the examples of nanomaterials for immunomodulation should be collected and added in the manuscript.
3. In Section 5.2 “Inorganic NPs present a higher chemical stability and a higher functionalization capacity”, more references could be cited to broaden the introduction.
https://doi.org/10.2147/IJN.S328767
